# Is It Time to Commit to a Process to Re-Evaluate Oncology Drugs? A Descriptive Analysis of Systemic Therapies for Solid Tumour Indications Reviewed in Canada from 2017 to 2021

**Sandeep Sehdev** [1] **and Alexandra Chambers** [2,*]

[1] Division of Medical Oncology, Department of Medicine, The Ottawa Hospital Cancer Centre and The University of Ottawa, Ottawa, ON K1N 6N5, Canada; sehdev@toh.ca
[2] Novartis Pharmaceuticals Inc. Canada, Dorval, ON H9S 1A9, Canada
[*] Correspondence: alexandra.chambers@novartis.com

**Abstract:** We undertook an analysis of the Canadian Agency for Drugs and Technologies in Health (CADTH)'s health technology assessments (HTAs) of systemic therapies for solid tumour indications to determine if a mechanism to re-evaluate HTA decisions is needed based on the level of certainty supporting the original recommendation. To measure the certainty in the evidence, we analysed if: (1) overall survival (OS) was the primary endpoint in the pivotal trial, (2) median OS was available at the time of the recommendation, and (3) the expert review committee explicitly identified gaps in the evidence. There were 96 drugs approved by Health Canada that met our eligibility criteria between 1 January 2017 and 31 October 2021. Median OS was not estimable at the time of the recommendation in 57% of the positive recommendations, and the uncertainty in the magnitude of clinical benefit was identified by the expert review committee in 21% of the positive recommendations. There is uncertainty at the time of the HTA recommendation for many drugs, and thus a need to implement a process to re-evaluate drugs in Canada to allow patients timely access to promising therapies while ensuring long-term value of therapies to patients and the healthcare system.

**Keywords:** health technology assessment; reassessment; drug access; drug reimbursement

## 1. Introduction

Every year, new therapies become available to Canadians through the provincial and territorial universal healthcare systems. As in many other countries, the processes of reviewing drugs for public reimbursement are complex [1,2]. In addition to being complex, Canadian processes are also static and do not have agile mechanisms to re-evaluate drugs once they are added to public formularies.

Drugs entering the Canadian market require regulatory approval from Health Canada. Health Canada has three review streams: standard review (300-day timeline), a priority review (180-day timeline) [3], and an accelerated review for drugs seeking a Notice of Compliance with conditions (NOC/c) (200-day timeline) [4]. When a drug receives an NOC/c approval rather than a Notice of Compliance (NOC), the manufacturer is required to collect evidence based on the conditions specified by Health Canada and provide them to Health Canada when available.

After regulatory approval is granted, a drug is submitted for health technology assessment (HTA) to review its value compared to existing therapies and make recommendations to the public drug plans on whether it should be reimbursed. Canada has two HTA agencies (Canadian Agency for Drugs and Technologies in Health (CADTH) and Institut National d'Excellence en Santé et Services Sociaux (INESSS, Quebec only)). The HTA agencies have expert committees that review the clinical benefit, economic value, patient perspectives, and implementation considerations for the drug in order to make recommendations for reimbursement. CADTH has an expert review committee specifically to review drugs

for cancer indications called the pan Canadian Oncology Drug Review (pCODR) Expert Review Committee (pERC). If a drug receives a positive HTA recommendation, the next step in the process is for the public drug plans to collectively negotiate a price for the drug with the pharmaceutical company through the pan Canadian Pharmaceutical Alliance (pCPA). The final step in the process is for each province, territory, or federal drug plan to make its own funding decision for the drug under review based on their budget and other priorities (Figure 1).

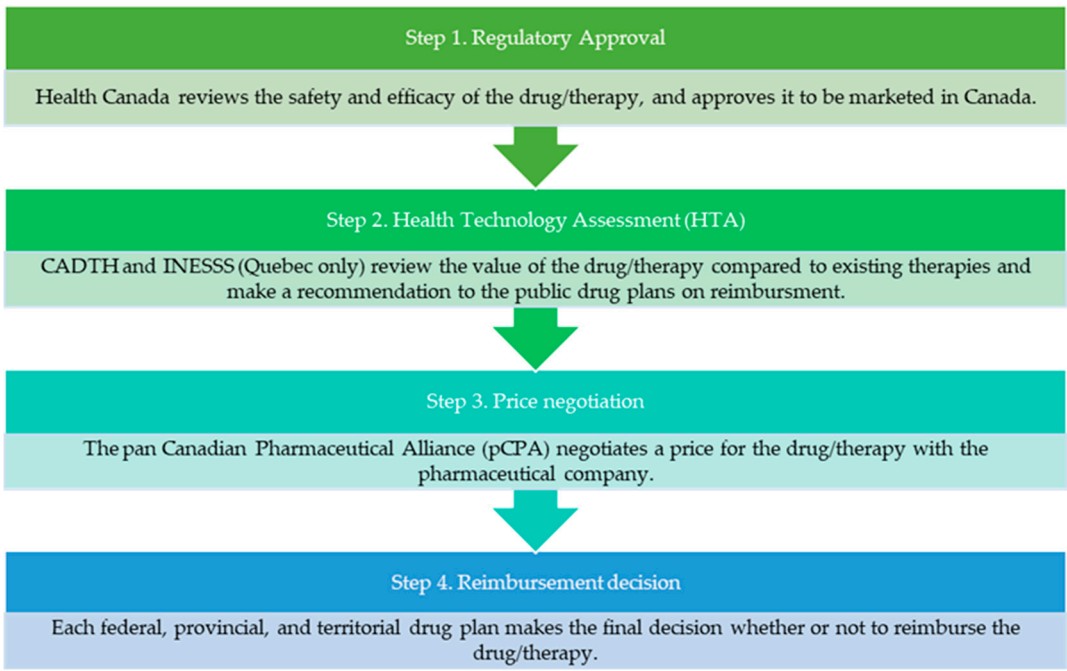

**Figure 1.** Drug reimbursement process in Canada. Abbreviations: CADTH, Canadian Agency for Drugs and Technologies in Health; INESSS, Institut National d'Excellence en Sante et Services Sociaux.

There is ongoing tension between population health and individual health, which results in tension among clinicians, patients, public payers, and the pharmaceutical industry regarding rigour and timeliness. There needs be a balance in how quickly a drug is added to a public formulary and confidence in the value that the drug offers patients and the healthcare system. Clinicians have obligations to society, the healthcare system, and most importantly, patients. Timely access is essential because patients could potentially benefit from these novel therapies, and some may die while waiting for them [5,6].

Currently, there is no clear mechanism to re-evaluate drugs once they are added to provincial formularies, the inert designs of which are barriers to re-evaluation. Once a drug is added to the formulary, it is not easily removed unless there is a clear signal that it is not safe. There has been discussion for several years that there should be a mechanism to re-evaluate drugs. This is sometimes referred to as a lifecycle approach [7], whereby a drug can be reviewed at different time points depending on the emergence of new evidence or new therapies entering the market. This concept of re-evaluation was prominent in CADTH's Strategic Plan for 2018–2021, where their priority was to "implement programs for reassessment and disinvestment" [7]. Similarly, the goal of the Canadian Real-World Evidence for Value of Cancer Drugs (CanREValue) collaboration is "to develop and test a framework for the generation and use of real-world evidence (RWE) of cancer drugs to enable (i) reassessment of cancer drugs by recommendation-makers and (ii) refinement of funding decisions or renegotiations/disinvestment by decision-makers/payers across Canada" [8].

There are several outcomes important to patients and clinicians in the treatment of cancer, including overall survival, disease-free survival, progression-free survival, quality

of life, and severity and frequency of adverse events [9,10]. Despite being the most objective, gold standard outcome, overall survival has limitations, including risks of confounding due to subsequent therapies and the duration of time it takes to measure overall survival compared to other meaningful outcomes [11].

We undertook this analysis of drugs for solid tumour indications to understand if a mechanism for re-evaluation is needed based on the certainty in the evidence available at the time of the HTA recommendations. To measure the certainty in the evidence, we analysed three key factors, if: (1) OS was the primary endpoint in the pivotal trial, (2) median OS from the pivotal trial was available at the time of the HTA recommendation, and (3) CADTH's pERC explicitly identified gaps in the evidence.

## 2. Methods

All drugs with completed new approvals or completed supplementary approvals were extracted from the Health Canada website from 1 January 2017 to 31 October 2021 [12]. The list of approvals was limited to drugs listed as "antineoplastic agents" on the Health Canada website to narrow the focus to drugs for oncology indications. The list was then further restricted to drugs for solid tumour indications. Supplementary approvals for dosing amendments were excluded. Biosimilars were also excluded from the list because CADTH does not review biosimilars. We extracted data from the Health Canada website on the pivotal study design, sample size, endpoints, and the type of regulatory review for the drugs that met the inclusion criteria.

We then assessed how many Health Canada-approved drugs had been reviewed by CADTH [13]. We extracted information from the CADTH website on the final recommendation and gaps in evidence identified by the pCODR Expert Review Committee (pERC) in their recommendations. For our analysis, we focused on CADTH recommendations because the majority of the provincial, territorial, and federal plans rely on CADTH recommendations to inform their decision-making.

Our descriptive analysis examined trends in regulatory approvals and HTA recommendations by tumour site, pivotal study design, and primary and secondary outcomes. In order to measure the certainty in the evidence at the time of the CADTH HTA recommendation, we extracted details on whether overall survival (OS) was measured and if OS was available at the time of the HTA recommendation. In addition, we reviewed each of pERC's recommendations to determine if they had explicitly identified gaps in the evidence. Until recently, pERC included a section in their recommendations called "Potential Next Steps for Stakeholders" which would outline factors that would need to be considered upon the introduction of the drug to the healthcare system. These 'next steps' could include references to pricing or administration considerations, and when applicable, pERC would identify gaps in the evidence not addressed in the HTA review. We analysed these 'next steps' to identify gaps in evidence flagged by pERC. The 'next steps' are less consistently reported for negative HTA recommendations because there is an implicit assumption that public reimbursement is unlikely.

We conducted an additional analysis to explore if there may be a signal to trigger a re-evaluation based on the European Society of Medical Oncology's Magnitude of Clinical Benefit Scale (ESMO-MCBS version 1.1) score. The MCBS was designed to assist in the prioritisation of medicines for cancer care [14]. The ESMO-MCBS assigns a score of 1–5 for drugs that are non-curative (5 being the highest score), and a score of A, B, or C for drugs that are curative (A being the highest). According to ESMO, a score of '4' or '5' in the non-curative setting or 'A' or 'B' in the curative setting is considered a substantial benefit, and lower scores are considered a non-substantial benefit [15]. We extracted the ESMO-MCBS score for each drug from the ESMO website [16].

We used the results of the descriptive analysis on the uncertainty at the time of the HTA recommendation as the basis to determine whether there is merit in having a mechanism to confirm the value of the therapy at a subsequent time point.

## 3. Findings

### 3.1. Overview of Systemic Therapies Approved by Health Canada

There were 96 drugs approved by Health Canada for new or supplementary indications that met our eligibility criteria between 1 January 2017 and 31 October 2021. The majority of the approvals were for lung cancer indications (*n* = 29), followed by gastrointestinal cancers (*n* = 17) and breast cancers (*n* = 16) (Figure 2). Notably, since 2019, there have been four Health Canada approvals for drugs for tumour agnostic indications. The full list of drugs included in our analysis are included in the Supplementary Material, Table S1.

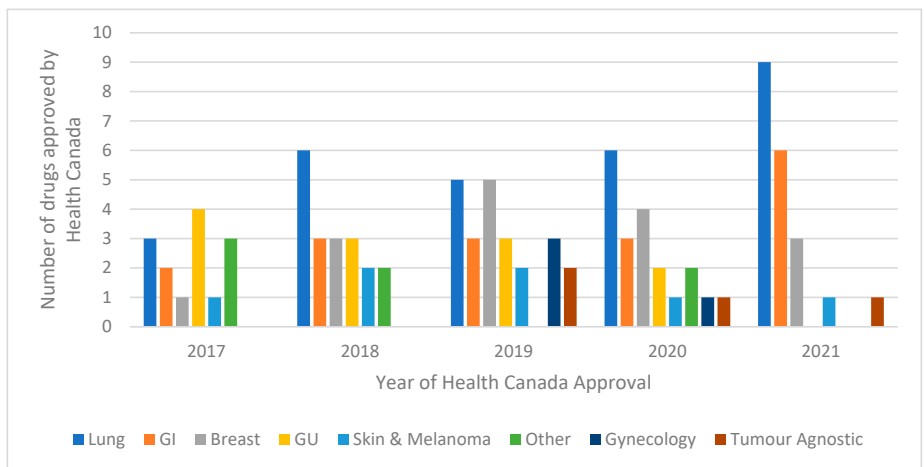

**Figure 2.** Number of drugs for solid tumour indications approved by Health Canada by year from January 2017 to October 2021. Abbreviations: GI, gastrointestinal; GU, genitourinary.

Most of the drugs approved by Health Canada have been submitted to CADTH for HTA review (73/96, 76%). However, nearly 1/4 of the Health Canada-approved drugs in our analysis have not been submitted to CADTH for an HTA recommendation. Of the drugs submitted to CADTH, about 70% (51/73) received a positive HTA recommendation. For the drugs in our analysis, CADTH issued 14 negative HTA recommendations, and as of 31 October 2021 there were 7 ongoing HTA reviews. One drug was withdrawn from the HTA review in 2019 (atezolizumab for triple-negative breast cancer). There are more drugs in the not submitted category for 2021 than for the other years, and this is likely due to the fact that some of these drugs will be submitted for HTA review in 2022 now that they have regulatory approval from Health Canada (Figure 3).

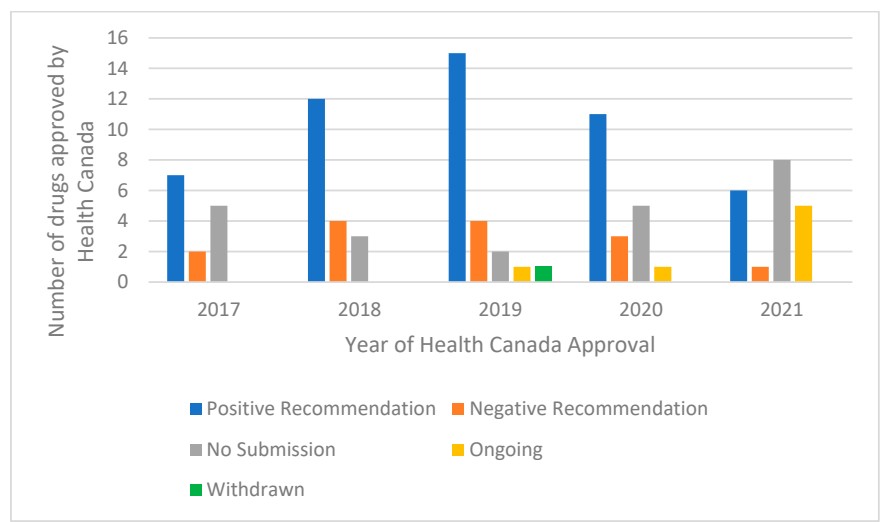

**Figure 3.** CADTH recommendation by year of Health Canada from 1 January 2017 to 31 October 2021.

About 46% of the drugs in our analysis followed the standard review stream at Health Canada (44/96), and there was nearly an even split of priority reviews and accelerated NOC/c reviews (25 and 27, respectively). Approvals based on the accelerated NOC/c review stream were less likely to be submitted to CADTH than approvals that followed the other streams (Figure 4).

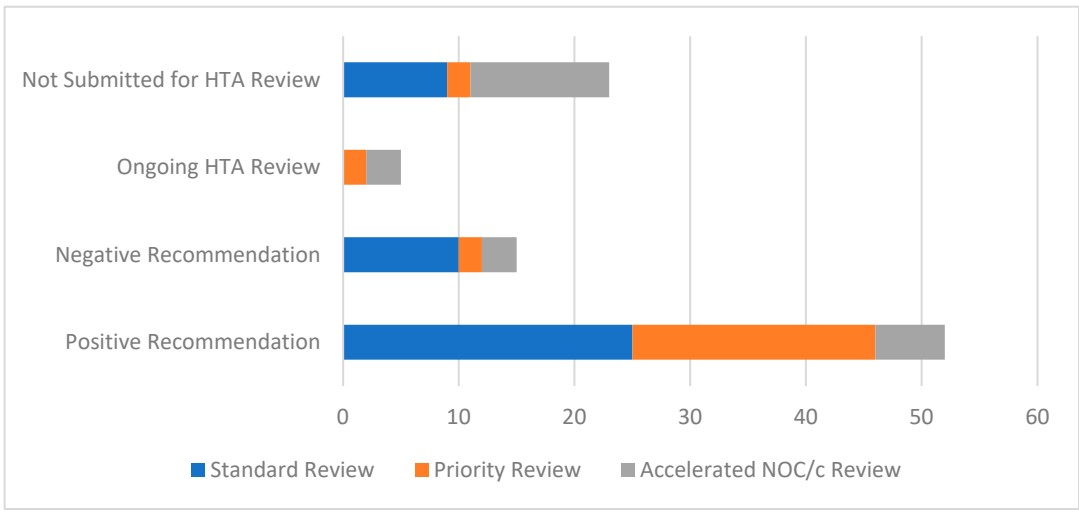

**Figure 4.** Status of CADTH review/recommendation by type of Health Canada regulatory review for drugs for solid tumour indications approved by Health Canada from 1 January 2017 to 31 October 2021. Abbreviations: NOC/c, Notice of Compliance with conditions.

While the majority of Health Canada approvals for drugs in our analysis were based on phase III randomised controlled trials (RCT) (66/96, 69%), about one-third of the Health Canada approvals were based on phase II trials (30/96, 31%). Of the 30 approvals based on phase II pivotal trials, 4 approvals were based on comparative phase II trials, while the remainder of the approvals were based on phase II non-comparative trials.

There were more positive HTA recommendations issued based on drugs where the pivotal trial was an RCT compared to drugs where the pivotal trial was a phase II study. However, there were still more positive HTA recommendations than negative HTA recommendations issued in drugs where the pivotal trial was a phase II study (8 positive recommendations vs. 6 negative recommendations).

*3.2. Uncertainty in the Evidence and the Time of the HTA Recommendation*

Overall survival was an endpoint in 82% of the pivotal trials for drugs with Health Canada approval; of these, OS was the primary endpoint in about 40% (31/78). Notably, all of the positive HTA recommendations were for drugs where the pivotal trials measured OS as a primary or secondary endpoint. There were no positive HTA recommendations where OS was not measured in the pivotal trial, and one negative HTA recommendation where OS was not measured. About half (12/23) of the studies for the drugs that were not submitted for an HTA review did not measure OS as an outcome (Figure 5).

Overall survival was not estimable at the time of the HTA recommendation in 41/73 (56%) where OS was an endpoint (either primary or secondary) of the pivotal trial. When analysing just the positive HTA recommendations, median OS was not estimable at the time of the HTA recommendation in 29/51 (57%) of the recommendations. Median OS was not estimable at the time of the HTA recommendation in a similar proportion of negative recommendations (8/14 (57%)) (Figure 6).

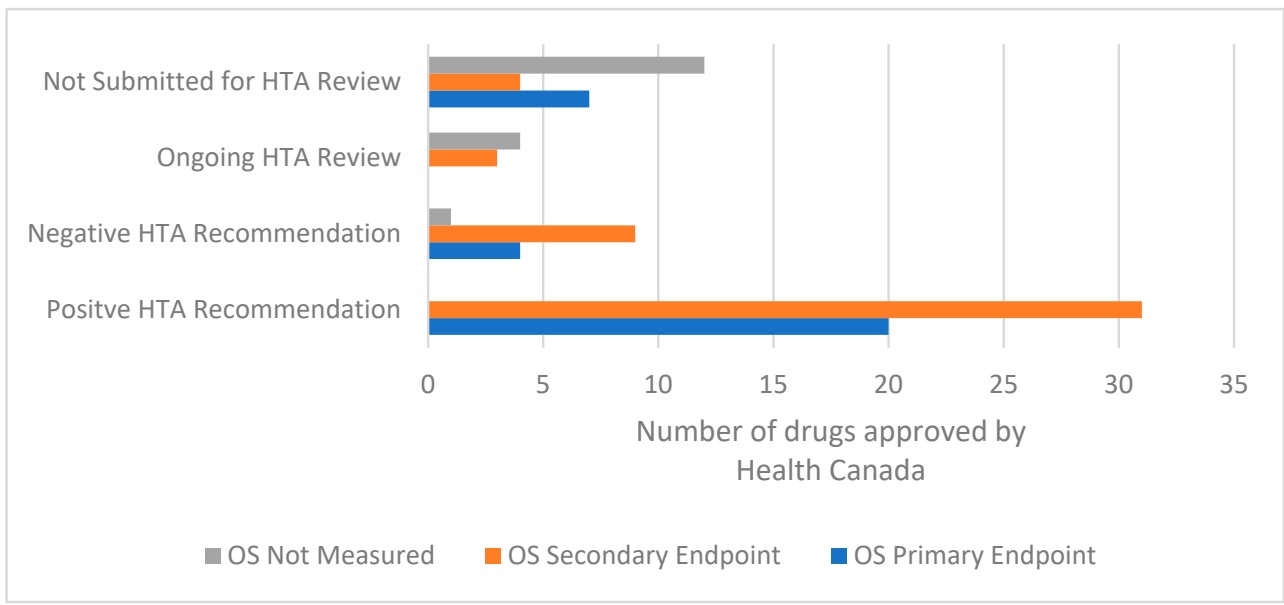

**Figure 5.** Comparison of overall survival as an endpoint and CADTH recommendation or status for drugs for solid tumour indications approved by Health Canada from 1 January 2017 to 31 October 2021. Abbreviations: HTA, health technology assessment; OS, overall survival.

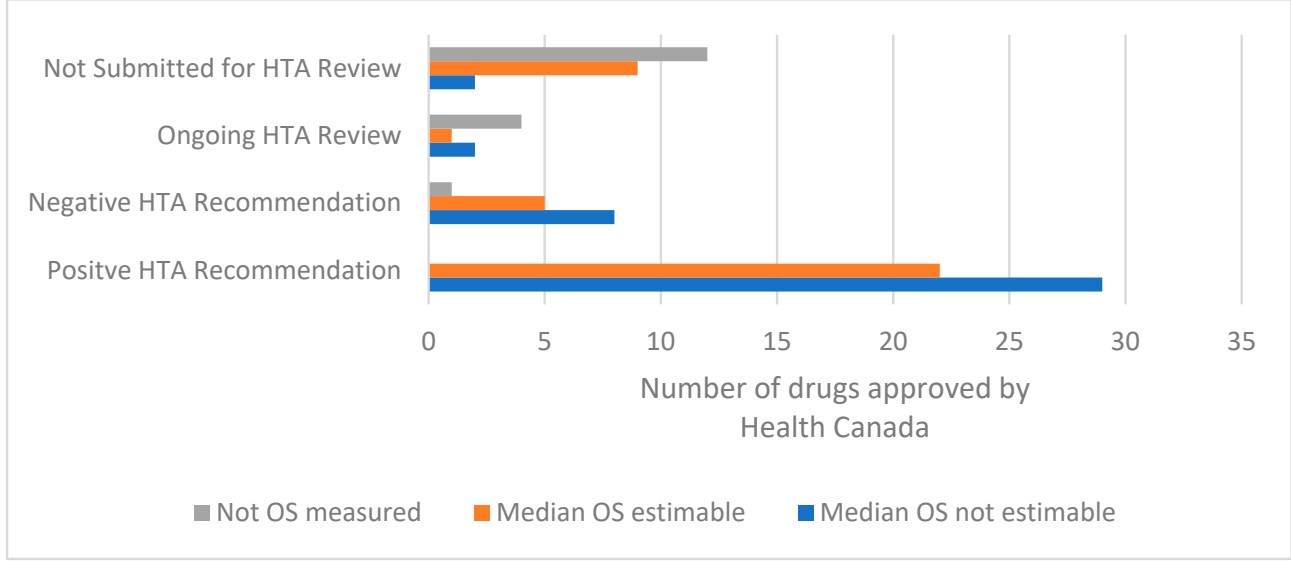

**Figure 6.** Availability of median overall survival results at the time of the HTA recommendation for drugs for solid tumour indications approved by Health Canada from 1 January 2017 to 31 October 2021. Abbreviations: HTA, health technology assessment; OS, overall survival.

When the "Potential Next Steps for Stakeholders" section was analysed in each of the positive HTA recommendations, 39 gaps in evidence were identified in 35 recommendations. Some recommendations had more than two gaps identified. We categorised the evidence gaps into three groups: (i) uncertainty in the magnitude of clinical benefit, (ii) uncertainty in sequencing the new drug with other currently available therapies, and (iii) uncertainty in treatment duration. The most common gap identified was uncertainty in sequencing with other available therapies. Uncertainty in the magnitude of clinical benefit compared to other relevant therapies was identified in 21% (11/51) of positive HTA recommendations.

Of the drugs with HTA recommendations, there were 15 drugs that met all 3 certainty criteria (15/73, 21%). That is, there were 15 drugs that used median OS as the primary

endpoint, OS was measurable at the time of the HTA recommendation, and there were no gaps in evidence identified by pERC. There were 11 positive HTA recommendations and 4 negative HTA recommendations that met all 3 certainty criteria. Conversely, there were 29/73 (40%) of the drugs with HTA recommendations that met none of the 3 certainty criteria. Of these, 18 resulted in positive HTA recommendations and 11 resulted in negative HTA recommendations (Figure 7).

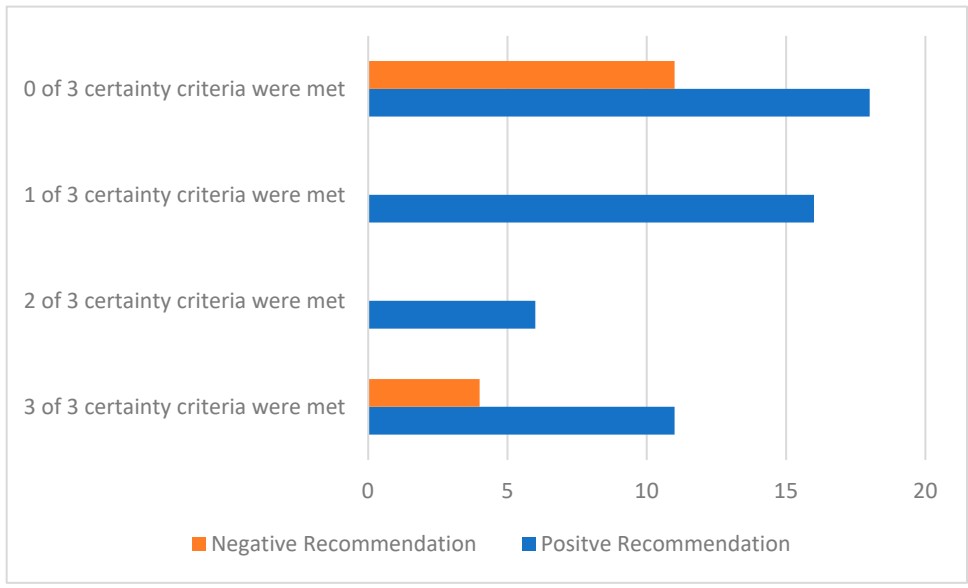

**Figure 7.** The number of positive and negative HTA recommendations by the number of certainty criteria that each drug review met. The certainty criteria are: (1) median OS as the primary endpoint, (2) OS was measurable at the time of the HTA recommendation, and (3) no gaps in evidence identified by pERC.

There were 14 negative HTA recommendations issued for the drugs included in our analysis. The rationale for the negative recommendations was largely due to uncertainty in the magnitude of clinical benefit. The pivotal trial was a non-comparative phase II study for five of these negative recommendations leading to uncertainty in clinical benefit compared to current standards of care. The pivotal trial was an RCT in seven of these recommendations, but the reimbursement request focused on a subgroup in four of these reviews, raising concerns in the confidence in the results. There were two reviews where the demonstrated lack of meaningful clinical benefit seemed to form the basis of the negative recommendation. In comparison, there were 12 reviews where the uncertainty in the magnitude of clinical benefit due to study design or immaturity of results led to a negative recommendation.

When we examined the ESMO Magnitude of Clinical Benefit Scale (MCBS) scores for the drugs in our analysis, we found that 76/96 (79%) were scored using the non-curative MCBS, 10/96 (10%) were scored using the curative MCBS, and 10/96 (10%) did not have an ESMO MCBS score on the ESMO website. About 37% (28/76) of the drugs scored with the non-curative MCBS scored a '4' or '5'. There was one drug that was assessed as a '4' but received a negative CADTH HTA recommendation. This recommendation was for lenvatinib with everolimus for renal cell carcinoma [17]. CADTH's pERC issued a negative recommendation because they were "not satisfied that there is a net clinical benefit of lenvatinib . . . " based on a randomised phase II trial. However, the committee noted that they would consider the drug again if "comparative efficacy data important to decision-making were provided". There were 2 drugs that scored '5' on the ESMO-MCBS that have not been submitted for an HTA review. Both of these drugs were for PD-L1-positive first-line metastatic lung cancer (pembrolizumab and atezolizumab) (Figure 8).

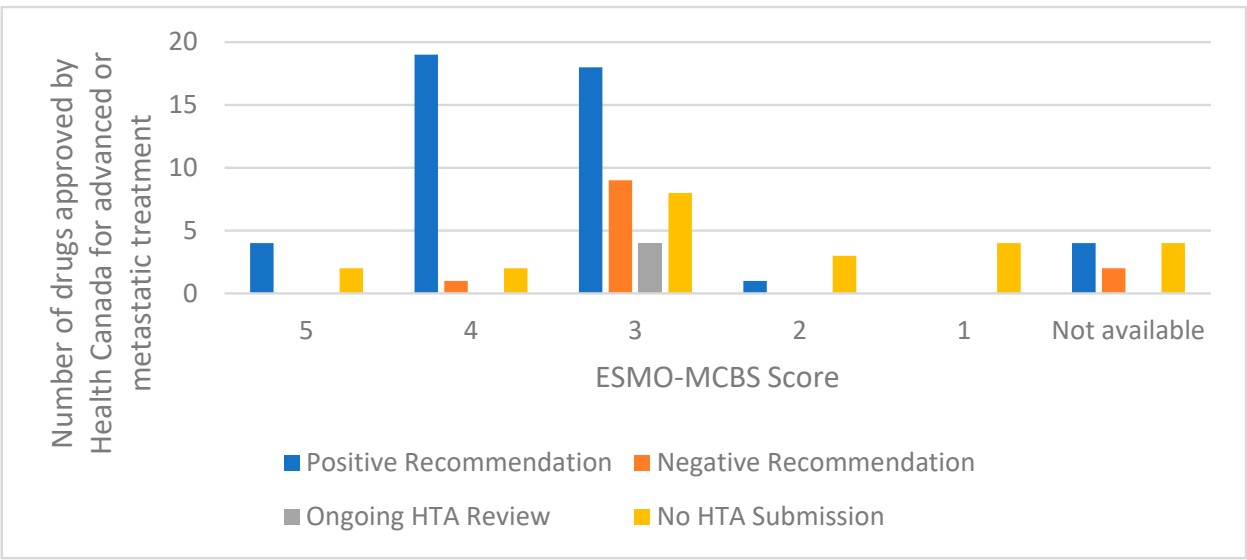

**Figure 8.** HTA recommendation or status based on the ESMO-MCBS score for drugs in the analysis for non-curative treatment. Abbreviations: HTA, health technology assessment; ESMO-MCBS, European Society of Medical Oncology Magnitude of Clinical Benefit Scale.

Of the 10 drugs scored using the curative MCBS, 9 of 10 drugs received an 'A' score, and 1 received a score of 'C'. There were three negative HTA recommendations in this 'curative' group, all for breast cancer indications (two for adjuvant treatment received an 'A', and the other for neo-adjuvant treatment received a 'C').

## 4. Discussion

We performed an analysis of systemic therapies for solid tumour indications to understand if a mechanism for re-evaluation is needed based on the certainty in the evidence available at the time of the HTA recommendation. Our analysis focused on three key factors to measure the certainty in the evidence in the HTA recommendations, i.e., if: (1) OS was the primary endpoint in the pivotal trial, (2) median OS from the pivotal trial was available at the time of the HTA recommendation, and (3) CADTH's pERC explicitly identified gaps in the evidence.

At the time of the HTA recommendation, 21% (15/173) of drugs met all 3 certainty criteria, 8% (6/73) met 2 of 3 certainty criteria, 22% (16/73) met 1 of 3 criteria, and 40% (29/73) met none of the criteria. About 1/3 of the positive HTA recommendations did not meet any of the criteria, suggesting a high level of uncertainty in the recommendation. It is important that even though there is uncertainty in the evidence at the time of the HTA recommendations, there is a pathway for patients to receive therapies.

Nearly 3/4 of the negative HTA recommendations did not meet any of the certainty criteria. Based on these findings, we wonder if there should be a delineation between negative recommendations based on a definitive lack of clinical benefit vs. those with uncertainty in clinical benefit? The extent that there is considerable uncertainty at the time of both the positive and negative HTA recommendations highlights the need for opportunities for further evaluation at a subsequent time point to ensure that value is being met. It is important to strike the balance between timely access because there is an urgency for patients to receive therapies and confirmation of clinical value.

There is an awareness that knowledge evolution is important to clinical practice and optimal patient care, as was recently highlighted by Bayoumi and Laupacis [18]. However, the current drug review system requires a fundamental shift in how the drug review process is implemented. There is a recognised need for both faster approval of breakthrough drugs and formal reappraisal of long-term benefit and real-world performance. Other countries have recognised the need and have implemented mechanisms to re-evaluate

drugs, sometimes integrated with risk-sharing financial agreements to allow for early approval and access. In the UK, they have the Cancer Drugs Fund (CDF) as a response to a report commissioned in the UK that demonstrated that they ranked poorly in terms of timely access to cancer drugs [19]. Despite higher drug expenditures, Canada scored worse than the UK in that analysis, better only than New Zealand. The CDF provides access to new treatments via managed access agreements while further evidence is collected to address clinical uncertainty [20]. France has created a temporary authorisation for use (ATU) program to enable patients to receive therapies with a high clinical benefit before market authorisation. They have been able to demonstrate that those granted ATU have all proceeded to receive regulatory approval [21]. The Netherlands introduced a conditional funding mechanism in 2005 for drugs that met certain criteria [22]. Italy's national medicines agency (AIFA) has various mechanisms in place for earlier access to promising therapies [23]. Based on 25-year data from the U.S. FDA's Accelerated Approval program, only 5% of indications for malignant haematology or oncology products have been withdrawn from the market because the confirmatory trials did not verify clinical benefit [24]. These mechanisms are not without their challenges [25]; however, their value is recognised [22]. Additionally, these processes for re-evaluation have been refined over time. For instance, the UK overhauled its CDF in 2016 [26].

Canada is well-positioned to develop its own process to re-evaluate drugs. There are rich sources of data in Canada to draw upon and strengthen to enable the generation of evidence to demonstrate value, such as the BC Cancer Registry [27] and the Alberta Cancer Registry [28]. An investment in Canada's data infrastructure would allow for the timely collection of quality real-world evidence, crossing provincial boundaries, assessing important variables, filling key data gaps, and together with international data, informing re-evaluation decisions. As Canada considers the development of a process to re-evaluate drugs, it would be beneficial to consider surveying patients and clinicians in Canada on their expectations and concerns about drug re-evaluation. For instance, if informed upfront that a drug was undergoing additional evidence generation, would patients and clinicians accept a decision to alter reimbursement (e.g., change starting/stopping criteria, duration of therapy, sequencing, listing status) on the drug once more evidence was available?

Canada already has some experience with re-evaluations. In 2011, Ontario established their Evidence Building Program to provide time-limited coverage for drugs while data were collected [29]. The first drug evaluated through the EBP was trastuzumab in combination with chemotherapy for the adjuvant treatment of small, early HER2-positive breast cancer, which allowed clinicians and patients access to this treatment while Cancer Care Ontario collected additional data. In 2019, Quebec's HTA agency, INESSS, issued a recommendation for a CAR T-cell therapy (Kymriah, tisagenlecleucel) whereby the HTA expert committee requested to re-evaluate the evidence in three years due to the uncertainty in the longer-term efficacy [30]. In making this recommendation, INESSS signalled a willingness to develop a framework for a re-evaluation process.

In addition, there is a precedent in Canada for accepting drugs based on conditions through Health Canada's Notice of Compliance with conditions (NOC/c) regulatory approvals. One potential starting point is to look at aligning Health Canada's existing process for conditional approvals (NOC/c) with conditional HTA approvals where there is promise of value but uncertainty [31]. Alternatively, the ESMO-MCBS may serve as a possible trigger for re-evaluation, perhaps in cases where the ESMO score does not align with the HTA recommendation. In 2018, CADTH's pERC issued a handful of recommendations that suggested that provincial drug programs consider "time-limited reimbursement" until more evidence was available to conduct a re-evaluation [32–35].

One of CADTH's strategic priorities is to "implement programs for reassessment and disinvestment" [7]. Similarly, the goal of the Canadian Real-World Evidence for Value of Cancer Drugs (CanREValue) collaboration is "to develop and test a framework for the generation and use of real-world evidence (RWE) of cancer drugs to enable (i) reassessment of cancer drugs by recommendation-makers and (ii) refinement of funding decisions or

renegotiations/disinvestment by decision-makers/payers across Canada" [8]. In 2020, the International Network of Agencies for Health Technology Assessment (INAHTA) and Health Technology Assessment International (HTAi) developed a new internationally accepted definition of HTA that states that "HTA is a multidisciplinary process that uses explicit methods to determine the value of a health technology at different points in its lifecycle" [36]. In order to measure the value at different points in the lifecycle, there needs to be a mechanism for re-evaluation. Through these global and Canadian initiatives, there appears to be an acknowledgement that a process for re-evaluation should be explored. By demonstrating that many HTA recommendations are made when there is still uncertainty in the clinical evidence, we hope that there will be a recognition that re-evaluation should be prioritised.

In recent years, there has been an increase in global regulatory alignment through Project ORBIS [37] with the U.S. FDA and other regulatory agencies through the Access Consortium [38]. These collaborations are welcomed because they are designed to improve efficiency and timely access for patients. Perhaps similar collaborations can be explored amongst HTA agencies to seek similar efficiencies to enable re-evaluations. Further, at the "front end" of initial approval and access, as part of governments' societal obligations to patients, performance standards might be established defining maximum reasonable timeframes from Health Canada approvals to provincial reimbursement decisions. As in other similar jurisdictions, approvals should not only be anchored to OS and phase III data but consider valuable phase II trials, international guidelines, and the totality of evidence.

Our analysis has some limitations. Our analysis looked for Health Canada approvals up to the end of 31 October 2021, potentially limiting the possibility of finding a subsequent CADTH HTA review. The more recent Health Canada approvals are less likely to have a CADTH recommendation. However, CADTH does allow for submissions to be made while Health Canada is conducting their review (parallel review option), at the very least a HTA review could be ongoing at the time of Health Canada approval. In 2021, CADTH changed their HTA recommendation template and no longer included a section for "Potential Next Steps for Stakeholders". This is a limitation because we relied on this section to identify the evidence gaps flagged by pERC. In addition, a limitation to the 'next steps' is that they may not have been included consistently in all HTA recommendations. The 'next steps' are dependent on the deliberations of pERC. The 'next steps' are less consistently reported for negative HTA recommendations because there is an implicit assumption that public reimbursement is unlikely. Our analysis focused only on the endpoint of overall survival because it is considered the gold standard endpoint for clinical trials. There are many other important outcomes to patients beyond survival that were not analysed. Surrogate endpoints such as progression-free survival or disease-free survival were the most common primary endpoints in our analysis. This is expected because the utility of OS may be limited due to a few factors. Surrogate outcomes such as PFS or DFS may be less time-consuming to generate compared to OS, and these surrogate outcomes also have the potential to contribute information about quality of life and treatment failure. Nonetheless, we looked primarily at median OS because it is often considered the 'gold standard' for trial outcomes in oncology [9]. Finally, the timing of the ESMO assessment with their MCBS and the CADTH HTA recommendation is unclear. If the ESMO assessment is conducted after the HTA recommendation, there may be more evidence available at the time of their assessment, or vice versa, leading to variation in ESMO score vs. CADTH recommendation.

In the future, it would be worthwhile to conduct a similar analysis on drugs for haematology indications to evaluate if there are any common themes with these therapies. In addition, it would be important to explore how the cost-effectiveness analyses and the timelines for reimbursement relate to the uncertainty in the evidence. Finally, a future analysis could take a more comprehensive exploration into other endpoints and how it relates to drug reimbursement in Canada.

### 5. Conclusions

There is frequently some degree of uncertainty in the evidence at the time of the HTA recommendation. Therefore, there is a need to implement a process to re-evaluate drugs in Canada in order to balance the urgent need for patients to access therapies and the sustainability of the healthcare system. Through broad stakeholder collaboration and a mindset of continuous improvement, Canada could successfully enable a mechanism to allow patients earlier access to promising therapies while ensuring long-term value of therapies to patients and the healthcare system.

**Supplementary Materials:** The following supporting information can be downloaded at: https://www.mdpi.com/article/10.3390/curroncol29030156/s1, Table S1: List of all drugs for solid tumour indications approved by Health Canada from 1 January 2017 to 31 October 2021.

**Author Contributions:** Conceptualisation, S.S. and A.C.; writing—original draft preparation, A.C.; writing—review and editing, S.S. All authors have read and agreed to the published version of the manuscript.

**Funding:** This research was funded by Novartis Pharmaceuticals Inc. Canada.

**Institutional Review Board Statement:** Not applicable.

**Informed Consent Statement:** Not applicable.

**Data Availability Statement:** All data used in this study were from public sources.

**Acknowledgments:** We would like to thank Louise Binder and Ruth Pritchard for their useful comments on the analysis.

**Conflicts of Interest:** A.C. is an employee of Novartis Pharmaceuticals Inc. Canada and former employee of CADTH. S.S. has no conflict of interest to declare relevant to the topic.

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
