# Peer review of "Is It Time to Commit to a Process to Re-Evaluate Oncology Drugs? A Descriptive Analysis of Systemic Therapies for Solid Tumour Indications Reviewed in Canada from 2017 to 2021"

_curroncol, doi:10.3390/curroncol29030156_

Round 1
Reviewer 1 Report
The goal of the paper is clearly stated (We did an analysis of drugs for solid tumour indications to understand if a mechanism for re-evaluation is needed based on the certainty in the evidence available at the time of the HTA recommendation- page 8) and the conclusion reached is also clearly stated (There is a need to implement a process to re-evaluate drugs in Canada similar to other countries- page 10). The overarching issue I have with the paper is that the conclusion reached is not based on the research findings. The research findings are descriptive in nature and there is no explanation in the methods of how the descriptive data will be used to come to a conclusion. The lack of linkage between findings and conclusion is illustrated by the fact that the only argument provided to justify the conclusion is the fact that "Other countries have implemented mechanisms to re-evaluate drugs" (page 9). The conclusion reached is not due to the fact that the proportion of HTA recommendations where the OS was not the primary endpoint was below a given threshold, for example. Furthermore, even if there was a direct linkage between the conclusion reached and the findings, I would expect that a decision on the need for a re-evaluation process would be more about the difference between expected performance and observed performance which is not addressed in this research.
Reviewer 2 Report
- How consistently reported is the "Potential Next Steps for Stakeholders" given that the majority of the work carried out for pCODR was done by contractors?
- I would find Figure 2 more interesting if presented by year. So for 2017, how many were lung, GI, breast and then other? The colored bar chart also makes it hard to read the individual contributions of each year.
- Would add the 'n' to Figure 3.
- Figure 4 caption: "...by Year of Health Canada approval..."? Why so many that have no submission in 2021, is this like due to the fact they were submit in 2022?
- Add the name in line 159 and re-phrase to say "of these, OS was the primary endpoint".
- When discussing "Potential Next Steps for Stakeholders", it is not clear if the 39 recommendations were identified across all reviews or only in 39 (or less) reviews.
- I feel the discussion is missing an important piece on what it would mean for patients and the healthcare system if a re-evaluation process were instituted (based on experience in other jurisdictions). Would it mean ensuring only the most effective drugs are reimbursed? Would there be greater efficiency of use of resources? Would it encourage submitters to think beyond "past the post" recommendations?
Reviewer 3 Report
The study is absolutely outstanding, aimed to point out if the (complex) Canadian rules for drug for solid tumors approval could/should be modified, in particular in terms of time for clinical availability, of need/utility over time (in absence of actual re-evaluation).
According to CeoworldMagazine, in 2021 Canada was 23rd among the best Healthcare Systems in the World, were USA was 30th and Italy 37th. The “Top Ten” were: South Korea; Taiwan; Denmark; Austria; Japan; Australia; France; Spain; Belgium and United Kingdom (https://ceoworld.biz/2021/04/27/revealed-countries-with-the-best-health-care-systems, accessed Feb 4th 2022)
The study is absolutely outstanding, aimed to point out if the (complex) Canadian rules for drug for solid tumors approval could/should be modified, in particular in terms of time for clinical availability, of need/utility over time (in absence of actual re-evaluation).
Methods are reliable and well described.
Results (Findings)
96 anticancer drugs were approved between January 2017 and October 2021 and both timeline and results of the further steps for final decision are well presented both in text and in Fig 3 to 5.
Two thirds (69%) of approvals were based on phase III randomized controlled clinical trials but one third on Phase II studies and positive reccomandations are more or less the same (Fig 6).
Important “gaps” were indentified in the successive step of the process.
When comparing the approval results with the ESMO Magnitude of Clinical Benefit Scale (MCBS) score, it is interesting the complexity of the procedure and discordance (as shown in Fig 10)
Discussion
Authors stress the fact that median Overall Survival , primary endpoint of the majority of trials, was not reported at the rime of reccomandation (step 2 of Fig 1). Absolutely agree with Authors that it is important for the rapid (?) availability of the drug but it is also important re-evaluate the role of the drug according to final results of the study
Reviewer 4 Report
This is a descriptive analysis of HC approved drugs for solid tumours submitted to CADTH for HTA. The authors sought evidence to support a mechanism to re-evaluate HTA decision by examining the level of certainty supporting the original recommendation. Level of certainty was determined by (1) whether or not OS was a primary endpoint in the pivotal clinical trial, (2) if mOS was available at the time of the HTA recommendation, and (3) whether the ERC explicitly identified gaps in the evidence.
There were 96 HC approved drugs for solid tumours (new or supplementary indication) Jan 1, 2017 to Oct 31, 2021. This excludes HC approvals for new dosing/schedule and biosimilars). 73 of these drugs were submitted for CADTH HTA.
Unfortunately, the level of certainty data for the drugs with actual CADTH recommendations is buried in tables and the presentation seems to be focused on the overall HC group. If evidence to support a mechanism to re-evaluate the HTA decision is being sought based on these criteria, the results for the drugs with actual HTA recommendations needs to be emphasized and still broken down for those with negative vs positive recommendation. In addition to bringing out the certainty evidence for the drugs with actual CADTH recommendations, the percentage of these drugs meeting all 3 certainty criteria, 2, 1, none could have been shown again for those with negative vs positive recommendation.
Discussion and conclusions should be based on the data sought for the drugs with actual CADTH recommendations.
Although the authors sought to find evidence to support a mechanism to re-evaluate HTA decisions, the discussion veers into an argument for more rapid approval of drugs (paragraph 2 of discussion). The authors use the FDA AA experience to support this argument (only 5% of of heme/onc drugs withdrawn over 25 years because confirmatory trials did not verify clinical benefit). It would be important for the reader to understand how many of the FDA AA drugs never were funded in Canada and how the FDA AA bar for verification of clinical benefit compares with CADTH). It is possible with such details that the FDA AA experience could be an argument against Canada pursuing a more rapid approval process. Is there literature to support oncology outcomes with the FDA AA in place are better than in Canada or vice versa?
Other suggestions and queries:
- Title: Is it time to commit to a process to re-evaluate oncology drugs? A descriptive analysis of systemic therapies for solid tumour indications… (line 3)
- Delete: Our analysis focused only on the endpoint of overall survival…(lines 95-97) Belongs in discussion.
- Paragraph starting on line 104: Part of what CADTH attempts to do is the same as what ESMO is doing with magnitude of clinical benefit scale. I would only argue that the ESMO-MCBS score could be looked at to potentially trigger re-evaluation if completed or updated after CADTH recommendation due to updated analyses or new evidence. In this case, why not just look for updated analyses or new evidence presented or published within a certain time period following the CADTH recommendation?
- Paragraph starting on line 122: Consider examining positive/negative CADTH recommendation amongst those submissions that were completed and not withdrawn).
- Paragraph starting on line 151: This is related to above comment. Please mimic results as presented in above paragraph but for just those with an actual CADTH recommendation. Please be consistent with tense (this paragraph uses present tense).
- Paragraph starting on line 159: Again, please present results in text for those with an actual CADTH recommendation.
- Paragraph starting on line 170: Again, please present results in text for those with an actual HTA recommendation.
- Paragraph starting line 178: Please express fraction (percent) for sequencing, MCB and duration for the overall group and again, for those with an actual HTA recommendation.
- Paragraph starting on line 190: Is exploration of ESMO-MCBS score valid in absence of temporal relationship (i.e. ESMO-MCBS score being published or updated after CADTH recommendation)?
- Consider delete Figure 2: HC approval by tumour group nicely described in paragraph above. Visualizing this in graph with added complexity by year does not add important information or help the reader understand the data.
- Consider delete Figure 3: Status of drugs submitted to HC described in paragraph above. Pie chart does not add information or help reader understand data.
- Consider delete Figure 4: CADTH recommendation by year not discussed in text and does not add to manuscript.
- Consider delete Figure 6: With revisions, this data can be best described in text for overall study period. Visualizing results by year is not helpful.
- Figure 9: Suggest redoing this figure in a way that is consistent with presentation of other 2 certainty outcomes – Figures 7 & 8.
Reviewer 5 Report
This commentary paper looked at the mecanism of drug for solid tumor approval in Canada (except Québec) to determine if a mechanism for re-evaluation of HTA decision is needed based on the level of uncertainty.
The measure of certainty was mostly based on OS.
The authors have well presented the point that drug approval is for now, a static process and not a continuum of continuous re-valuation of benefits of drugs.
Results are well presented as well as the discussion.
My only comment relates to the discussion of endpoint. OS is cited has the prime outcome. However, many trials in oncology have either DFS or iDFS as endpoint or PFS. Given the "long" time before the event-death-happens, surrogate outcomes are often used. Trials are powered for the primary endpoint and if not OS, not powered for OS. It would be helpful for the reader to discuss the choice of endpoint and its impact on the process of drug approval.
Round 2
Author Response
Thank you very much for your additional review and feedback. We appreciate the thoughtful comments.
To strengthen the statement regarding the processes in other countries, we have added a few sentences in the discussion to highlight that the definition for HTA was recently updated and internationally accepted by INAHTA and HTAi to state that “HTA is a multidisciplinary process that uses explicit methods to determine the value of a health technology at different points in its lifecycle.” This addition of "different points in its lifecycle" adds to the need and global acknowledgement for re-evaluation.
Despite this addition, we agreed with your comment regarding the conclusion and removed the phrase "similar to other countries".
Finally, thank you for identifying the need for the edits to lines 87 and 196. The sentences have been revised.
Your considerable feedback has been very welcomed and appreciated, we believe it has genuinely improved our manuscript.